# Zika Virus Non-Structural Protein 1 Antigen-Capture Immunoassay

**DOI:** 10.3390/v13091771

**Published:** 2021-09-05

**Authors:** Brandon J. Beddingfield, Jessica N. Hartnett, Russell B. Wilson, Peter C. Kulakosky, Kristian G. Andersen, Refugio Robles-Sikisaka, Nathan D. Grubaugh, Argelia Aybar, Maria-Zunilla Nunez, Cesar D. Fermin, Robert F. Garry

**Affiliations:** 1Department of Microbiology and Immunology, School of Medicine, Tulane University, New Orleans, LA 70112, USA; bbedding@tulane.edu (B.J.B.); jessica.n.hartnett@gmail.com (J.N.H.); 2Autoimmune Technologies, Limited Liability Company, New Orleans, LA 70112, USA; rbw@autoimmune.com (R.B.W.); pck@autoimmune.com (P.C.K.); 3Department of Immunology and Microbial Science, Scripps Research, La Jolla, CA 92037, USA; kga1978@gmail.com (K.G.A.); rrobles@scripps.edu (R.R.-S.); nathan.grubaugh@yale.edu (N.D.G.); 4Scripps Translational Science Institute, La Jolla, CA 92037, USA; 5Department of Integrative Structural and Computational Biology, Scripps Research, La Jolla, CA 92037, USA; 6MediPath Instituto de Patologia Molecular, Universidad Tecnológica de Santiago (UTESA), Santiago 51000, Dominican Republic; argelia.aybar@gmail.com; 7Centro de Investigaciones Biomédicas y Clínicas (CINBIOCLI), Pontificia Universidad Católica Madre y Maestra (PUCMM), Santiago 51034, Dominican Republic; nunez.zunilda@gmail.com; 8Instituto de Innovacion Biotecnologia e Industria (IIBI), Santo Domingo 10135, Dominican Republic; cedafs@icloud.com; 9Ministerio de Salud Publica (MSP), Santo Domingo 10514, Dominican Republic; 10Zalgen Labs, Limited Liability Company, Germantown, MD 20876, USA

**Keywords:** Zika virus, non-structural protein 1, site-directed mutagenesis, polyclonal antibodies, antigen-capture ELISA

## Abstract

Infection with Zika virus (ZIKV), a member of the *Flavivirus* genus of the *Flaviviridae* family, typically results in mild self-limited illness, but severe neurological disease occurs in a limited subset of patients. In contrast, serious outcomes commonly occur in pregnancy that affect the developing fetus, including microcephaly and other major birth defects. The genetic similarity of ZIKV to other widespread flaviviruses, such as dengue virus (DENV), presents a challenge to the development of specific ZIKV diagnostic assays. Nonstructural protein 1 (NS1) is established for use in immunodiagnostic assays for flaviviruses. To address the cross-reactivity of ZIKV NS1 with proteins from other flaviviruses we used site-directed mutagenesis to modify putative epitopes. Goat polyclonal antibodies to variant ZIKV NS1 were affinity-purified to remove antibodies binding to the closely related NS1 protein of DENV. An antigen-capture ELISA configured with the affinity-purified polyclonal antibody showed a linear dynamic range between approximately 500 and 30 ng/mL, with a limit of detection of between 1.95 and 7.8 ng/mL. NS1 proteins from DENV, yellow fever virus, St. Louis encephalitis virus and West Nile virus showed significantly reduced reactivity in the ZIKV antigen-capture ELISA. Refinement of approaches similar to those employed here could lead to development of ZIKV-specific immunoassays suitable for use in areas where infections with related flaviviruses are common.

## 1. Introduction

Zika virus (ZIKV) is named after the Zika forest in Uganda, where it was first isolated from a sentinel monkey in 1947 [1], and shortly after from mosquitoes in the same area [2]. Follow-up serological studies revealed Zika to be widespread in Africa and Asia [1,3,4]. The first reported natural ZIKV infection in humans was reported in 1964, when a scientist was infected while isolating virus from mosquitos in Uganda [4,5]. The first known incidence of Zika outside Africa and Asia was in 2007 in Yap State, Micronesia [6], in what became the first large outbreak recorded [7]. A total of 73% of residents of Yap Island 3 years old or older were infected [8]. In 2013, cases of Zika began being reported in French Polynesia. A total of 8746 cases were estimated to have occurred during this outbreak [9]. Other ZIKV outbreaks included New Caledonia, Cook Islands and Easter Island [10]. 

Beginning in early 2015, cases of ZIKV infection were reported in Bahia, Brazil [11]. A second introduction of Zika into Brazil occurred at about the same time, in Natal [12]. Estimates were from 440,000 to 1.3 million cases [13], with an eventual estimate of over 500,000 cases of Zika [14] and 73% of the population exposed in some areas [15]. Other countries in the Americas also reported Zika cases during this time. Colombia reported over 100,000 suspected cases [16], with a total of 27 countries in the Americas reporting cases, including Mexico, Guatemala, the Dominican Republic and Panama [17]. Eventually, Zika cases were reported in the continental United States, with multiple introductions of ZIKV into Florida [18], as well as Texas. These cases in Texas and Florida were locally acquired, indicating at least some level of circulation in local mosquito populations [19]. Afterward, ZIKV made its way to Cape Verde, Samoa, the Solomon Islands and the surrounding region [5,20]. In all, 87 countries were affected by the pandemic, with cases continuing after the large outbreak, including 30,000 cases in 2018 [14]. ZIKV has established itself as endemic to many more regions than it was prior to the outbreak and may continue to infect humans routinely into the future.

ZIKV infection usually results in a self-limited illness [21]. In symptomatic cases the most common symptom is a maculopapular rash, occurring in over 80% of cases [22], often on the face, torso and upper arms [23,24]. Another common symptom is arthralgia in about 62% of cases, mainly involving small joints [22,25], and occasionally larger joints as well [24,26]. Conjunctivitis is seen in roughly half of patients, with both eyes commonly being affected [22,27]. Other symptoms may be seen, such as myalgia, headache, nausea, vomiting, sore throat and cough [28]. If present, symptoms resolve within a week and are generally not fatal [29], provided there is not a comorbidity, such as sickle cell anemia [30]. Before 2013, no significant complications were reported from infections with ZIKV [4]. However, it was later determined that ZIKV is capable of establishing persistent infections, crossing placental and neuronal barriers, and damaging neurons [31]. During the French Polynesia outbreak, multiple cases of Guillain–Barré Syndrome (GBS) were reported [5,32,33]. GBS is a neurological disorder presenting as a rapid ascending paralysis that results in respiratory failure [34]. Another complication seen from ZIKV infections, microcephaly, occurs in a developing fetus in infected pregnant women. Defined as a neurological malformation resulting in decreased head circumference due to death of neural progenitor cells, microcephaly has been implicated as a result of ZIKV infections in eight individuals during the French Polynesia outbreak [35]. The outbreak in Brazil resulted in many more reports of congenital microcephaly, prompting the World Health Organization to declare a Public Health Emergency of International Concern on 1 February 2016 [36]. Other complications include conditions such as optic neuropathy, uveitis and congenital glaucoma resulting in loss of vision [37,38]. 

Antigen-capture diagnostic assays targeting NS1 have been utilized in the past for dengue virus (DENV) infection, with NS1 being established as an early biomarker for flavivirus infection [39,40,41,42]. The flavivirus non-structural protein 1 (NS1) functions in genome replication as an intracellular dimer and in immune system evasion as a secreted hexamer. Assays are often in Enzyme Linked Immunosorbent Assay (ELISA) format using monoclonal antibodies (mAbs) to overcome the genetic similarity among flaviviruses [43,44,45,46,47]. One of the assays commonly employed for the early diagnosis of dengue is the InBios DENV Detect NS1 ELISA Kit. It has a reported sensitivity of 86.8% and specificity of 97.8% [48]. One potential issue may be the lack of sensitivity in secondary dengue cases, with one study indicating 100% sensitivity in primary infections, and 10% sensitivity in secondary infections [41]. This is likely due to circulating anti-NS1 antibodies present during secondary infections [49]. This demonstrated success with NS1-based detection of DENV infection offers an attractive option for detection of infection with other flaviviruses. 

Here, we describe development of an NS1 capture diagnostic assay that utilizes affinity-purified polyclonal antibodies (pAbs) generated against recombinant ZIKV NS1 containing mutations in potentially cross-reactive epitopes. Reduced cross-reactivity to NS1 proteins of other commonly circulating flaviviruses was observed. The antigen-capture ELISA appears to be sensitive and specific, with little to no interference from antibodies generated against NS1 from the early immune response in patients. 

## 2. Materials and Methods

### 2.1. Mutagenesis of Zika Recombinant NS1 Protein

Site-directed mutagenesis was performed to generate mutated recombinant NS1 (rNS1). Primers were designed to mutate residues in regions of NS1 with high sequence similarity to other members of the *Flaviviridae* family (Appendix A). PCR was performed using a reaction consisting of 20 ng wild-type plasmid template, 0.5 µM forward and reverse primers, 1X Pfx AccuPrime Reaction Mix, 2.5 units AccuPrime Pfx polymerase (ThermoFisher, Waltham, MA, USA). Reaction conditions consisted of 1 cycle of initial denaturation of 95 °C for 5 min, 12 cycles of denaturation at 95 °C for 30 s, annealing at 56 °C for 1 min, and extension at 68 °C for 8 min. This was followed by a final annealing step of 56 °C for 1 min, and a final extension of 68 °C for 30 min. The PCR products were treated with DpnI (ThermoFisher, Waltham, MA, USA) for 1 h and purified using PureLink Quick PCR purification Kit (Invitrogen, Darmstadt, Germany). Purified products were then transformed into JM109 (Promega, Madison, WI, USA) or DH5α (Invitrogen, Waltham, MA, USA) cells via heat shock and plated for overnight incubation on LB agar containing 100 µg/mL carbenicillin at 37 °C. Colonies were picked and screened by PCR.

### 2.2. Cloning of NS1 into Expression Vector and Expression Screening

The full-length amplified NS1 genes for ZIKV and DENV type II, NCBI Accession Numbers LC002520 and KM204118, respectively, were ligated into the expression vector pET-45b(+) (Novagen, Madison, WI, USA) using the added restriction sites, resulting in an N-terminally hexahistidine-tagged NS1 protein under the control of the T7 promoter with minimal vector-encoded amino acids. The plasmid was transformed into Rosetta 2 (DE3) *E. coli* cells and plated on LB agar containing 100 µg/mL carbenicillin and incubated overnight at 37 °C. Rosetta 2 strains enhance expression of eukaryotic proteins that contain codons rarely used in *E. coli* by supplying tRNAs for 7 rare codons (AGA, AGG, AUA, CUA, GGA, CCC, and CGG) on a chloramphenicol-resistant plasmid. 

Colonies were screened by PCR amplification of gene insert followed by confirmation via restriction digestion using the enzymes used for insertion. Colonies confirmed by digestion were further confirmed by sequencing of the insert using primers for T7 promoter and T7 terminator sequences (GeneWiz, South Plainfield, NJ, USA). Colonies from a confirmed plasmid were screened for expression levels by plating on LB agar containing 100 µg/mL carbenicillin followed by growth in LB broth. At an OD_600_ of 0.7 in liquid culture, expression was induced using 1 mM IPTG and pelleted after a period of 3 h. Colonies were diluted to 100× the final OD_600_ with water and lysed with addition of Laemmli sample buffer (Bio-Rad, Hercules, CA, USA) to working concentration, followed by brief sonication. Lysed samples were analyzed by SDS-PAGE followed by total protein staining via Coomassie Blue G250.

### 2.3. Flavivirus Gene Synthesis and PCR Amplification

The full length NS1 genes from yellow fever virus (YFV), West Nile virus (WNV), and St. Louis Encephalitis virus (SLEV), NCBI Accession Numbers KF769016.1, DQ211652.1, and KX258461.1, respectively, were synthesized in pUC-IDT (Amp) (Integrated DNA Technologies, Coralville, IA, USA). The NS1 gene was then amplified using primers NS1 WT Forward (Appendix A) with the upstream restriction site and NS1 WT Reverse with the restriction site added downstream, as well as an *E. coli* STOP codon. Restriction sites differed based on whether each enzyme cut site was contained within the gene of interest; if so, a different enzyme was chosen. For SLEV, BamHI and HindIII were used, with YFV using MfeI and HindIII, and WNV using BamHI and NotI. PCR was performed using a reaction containing 1X Phusion polymerase (New England BioLabs, Ipswich, MA, USA), 200 µM dNTPs, 0.5 µM forward and reverse primers, 1X Phusion HF Buffer, and 3% DMSO using an ABI thermal cycler (Applied Biosystems, Foster City, CA, USA). Reaction conditions consisted of an initial denaturation at 98 °C for 30 s, followed by 35 cycles of denaturation at 98 °C for 10 s, annealing at 55 °C for 30 s, and extension at 72 °C for 30 s, with a final extension of 72 °C for 10 min. Inserts were confirmed by sequencing (GeneWiz, South Plainfield, NJ, USA). 

### 2.4. rNS1 Production and Solubilization

One-liter cultures of bacteria expressing recombinant NS1 were grown to an OD_600_ of 0.7 and induced with 1 mM IPTG. After induction for 4 h, cultures were harvested at 10,000× *g* for 10 min. Pellets were suspended in cell lysis buffer (10 mM Tris, 10 mM EDTA, 100 mM NaCl, 100 µg/mL Lysozyme, 1% Triton X-100, and protease inhibitor cocktail) for 1 h at 4 °C. Pellets were sonicated for 5 min at 40% power (10 s on/30 s off). The solution was centrifuged at 10,000× *g* for 40 min and resuspended in IB wash buffer (10 mM Tris, 200 mM NaCl, 1 M Urea, pH 6.0). The suspension was sonicated briefly and rocked for 1 h at 4 °C and centrifuged at 10,000× *g* for 40 min. The resulting inclusion body was resuspended in IB solubilization buffer (10 mM Tris, 500 mM NaCl, 100 mM NaH_2_PO_4_, 8 M Urea, and 10 mM Beta-mercaptoethanol, pH 8.0) overnight at 4 °C after a brief sonication. The solution was centrifuged at 3220× *g* for 30 min and used for rNS1purification.

### 2.5. Purification by Immobilized Metal Affinity Chromatography

The addition of the N-terminal His_6_ tag allowed for purification of rNS1 by immobilized metal affinity chromatography (IMAC) using HisPur Ni-NTA spin columns (Thermo Scientific, Waltham, MA, USA). The column was equilibrated with 2 bed volumes of IB solubilization buffer and the solubilized rNS1 was loaded onto the column. The protein was allowed to bind for 1 h at room temperature on a rotary shaker. Unbound protein was allowed to flow through via gravity, and the resin was washed with 10 bed volumes of wash buffer (500 mM NaCl, 100 mM NaH_2_PO_4_, 8 M Urea, 10 mM beta-mercaptoethanol, 20 mM imidazole, pH 8.0). Protein was eluted by passing 10 bed volumes of elution buffer (500 mM NaCl, 100 mM NaH_2_PO_4_, 8 M Urea, 10 mM beta-mercaptoethanol, 250 mM imidazole, pH 6.0) over the column. Fractions were collected for later SDS-PAGE analysis, and pure eluted fractions were pooled together for refolding.

### 2.6. Refolding of Recombinant NS1

Refolding of rNS1 of ZIKV, DENV, YFV, WNV and SLEV was accomplished by adjusting the concentration of the pooled protein to 100 µg/mL in IB solubilization buffer and added to a 10 MWCO dialysis flask (Thermo Scientific, Waltham, MA, USA). The protein was refolded at 4 °C over the course of 3 days in refolding buffer (50 mM Tris, 0.4 M L-arginine, 1 mM reduced glutathione, 0.1 mM oxidized glutathione, pH 8.0) at a 10-fold volume excess, changing refolding buffer once a day. Dialysis was subsequently performed into PBS (pH 7.4) using 3 changes of buffer, with a final overnight exchange.

### 2.7. Western Blot Analysis of Refolded Protein

Refolded rNS1 was analyzed by electrophoretic separation on SDS-PAGE gels and transferring the proteins from the gel to nitrocellulose using a XCell II Blot Module (Thermo Scientific, Waltham, MA, USA). The membranes were blocked for 1 h at room temperature with 5% nonfat dry milk in PBS and 0.05% Tween-20 (PBST). The blots were incubated with mouse anti-His_6_ monoclonal antibody (GE Healthcare, Chicago, IL, USA) or anti-ZIKV NS1 monoclonal antibody (GeneTex, Irvine, CA, USA) overnight at 4 °C with gentle rocking, followed by washing with PBST three times for 5 min each. Incubation with anti-mouse HRP antibodies followed for one hour at room temperature, and another wash with PBST three times for five minutes each, followed by development with Novex ECL chemiluminescent reagent (Invitrogen, Waltham, MA, USA).

### 2.8. Generation of ZIKV rNS1 Antiserum

Antiserum against ZIKV rNS1 proteins was generated via immunization of goats with wild-type NS1 and NS1 with alanine substitution mutations of amino acids 117–119 or 227–229 were each inoculated in goats. (ProSci, San Diego, CA, USA). Goats were immunized with 1.5 mg total of each protein four times at intervals of three weeks. A pre-immunization bleed was collected plus 4 post-immunization bleeds. Two goats were used for each protein, with the last bleed for each protein being used for experiments.

### 2.9. Generation of Polyclonal Antibodies against ZIKV NS1

Antiserum generated in goats against ZIKV NS1 117–119 were purified using affinity chromatography. CarboxyLink Coupling Resin (Thermo Scientific, USA) was coupled to 5–10 mg ZIKV NS1 117–119 in PBS using a 3-fold increase in the suggested amount of EDC to compensate for phosphate-containing buffer. Coupled resin was washed and the flow-through combined with the wash was analyzed at A280 to determine coupling efficiency by comparing this with the protein solution applied to the resin. Coupled resin was added to a column suitable for chromatography and stored at 4 °C in PBS + 0.02% sodium azide. Antiserum against rNS1 was diluted 1:1 with PBS and filtered through a 0.22 µm syringe filter before application to the column. Once applied to the column, the solution was allowed to flow via gravity until it reached the column exit. Fractions were collected in 1 mL aliquots for A280 analysis. Columns were washed using 10 resin bed volumes of PBS, followed by elution of bound antibodies with 0.1M glycine, pH 2.7. Elution fractions (1 mL each) were combined with 60 µL of 1M Tris-HCl, pH 9.0 for antibody affinity protection. The column was washed with at least 10 resin bed volumes of PBS to remove the glycine, followed by storage of the column in PBS + 0.5% sodium azide at 4 °C.

### 2.10. Purification of ZIKV Specific Antibodies

Anti-ZIKV NS1 pAbs were cross-adsorbed against DENV2 NS1 using affinity chromatography against DENV2 NS1 coupled to CarboxyLink resin via the same protocol as before. Antibodies eluted from the ZIKV NS1-coupled column were mixed 1:1 with PBS and applied to the DENV2 NS1-coupled column. The column was washed with 10 resin bed volumes of PBS, followed by elution of the bound fraction by 0.1 M glycine, pH 2.7. These fractions were combined for later experimental use as non-adsorbed ZIKV NS1 pAbs. The fractions that were collected in the flow-through and wash fractions were combined and applied to the DENV2 NS1-coupled column again. Flow-through and wash fractions were collected and combined to be used experimentally as cross-adsorbed ZIKV NS1 pAbs. Both of the cross-adsorbed and non-adsorbed solutions were concentrated via Amicon Ultra 30,000 MWCO spin filters (Millipore, Cork, Ireland). The pAbs were biotinylated via the EZ Link Sulfo-NHS Biotin Kit (Thermo Fisher, Waltham, MA, USA).

### 2.11. Antigen-Capture ELISA

Evaluation of ability of purified pAbs to bind NS1 proteins was performed via Antigen-capture ELISA. The affinity-purified pAbs were titrated to determine the optimal concentration to adsorb to the ELISA plates. A variety of coating conditions and ELISA plates were tested. Optimal conditions were found to be coating pAbs on Nunc Maxisorp 96-well plates overnight at 10 µg/mL in CBC at 4 °C. Wells were blocked with 1.5% BSA for four hours at room temperature before plates were dried overnight at room temperature and stored at 4 °C. Serum samples to be analyzed were diluted at 1:10 in sample diluent before addition to the antibody-coated plate and incubation at 37 °C for one hour. Plates were washed with PBS containing 0.05% Tween-20 (PBST) four times before addition of biotinylated-NS1 pAbs (5 µg/mL) for 30 min at room temperature. After another PBST wash, a 1:5000 solution of high sensitivity Streptavidin-HRP (Thermo Fisher, Waltham, MA, USA) was added and the plate was incubated at room temperature for 30 min. After another PBST wash, the plate was incubated with TMB substrate (Sigma, Neustadt, Germany) in the dark for 15 min. The reaction was then stopped with 0.36 N H_2_SO_4_ before reading the absorbance of each well at 450 nm. Standards were run on each plate consisting of ZIKV rNS1 diluted from 2000 ng/mL down to 0.49 ng/mL using 1:4 dilutions, plus a negative control consisting of normal human serum diluted 1:10 in sample diluent.

### 2.12. NS1-Antibody Complex Dissociation ELISA

Serum samples were incubated in an alkaline detergent solution in order to release antibodies from the antigen (NS1) to examine the possibility of antibody interference of antigen detection. Serum (55 µL), or soluble NS1 in the case of the standard curve, were mixed with 55 µL of dissociation solution (1 M Tris base, pH 10.5, 2% Triton X-100, and 150 mM NaCl) and incubated for 1.5 h at 37 °C. After incubation, the alkalinity was neutralized via addition of 20 µL of 2 M HCl and incubation for one hour at 37 °C. After incubation, 100 µL of 150 mM NaCl solution was added to dilute the detergent percentage for more efficient binding to the capture antibodies. After sample addition to the plate, the plates were incubated overnight at 4 °C and subjected to the rest of the above ELISA procedure the next day. Dissociated and non–dissociated samples were run on the same plate for comparison. Non-dissociated samples were diluted in pre-neutralized and diluted dissociation buffer. Standards consisted of soluble NS1 run side-by-side on the same plate using dissociated and non-dissociated conditions.

## 3. Results

### 3.1. Computational Analysis of Flavivirus NS1

Crystal structures for full-length, glycosylated NS1 from WNV and DENV have been reported (PDB: 5GS6, 5IY3). These structures define protein domains and potential immunoreactive sites (Figure 1). NS1 is a dimer with three major domains in each monomer (Figure 1A). Due to the sequence similarity between NS1 proteins of flaviviruses, the ZIKV NS1 is expected to be structurally similar to the NS1 proteins of WNV, DENV and other members of the flavivirus genus of the Flaviviridae. A small “β-roll” dimerization domain (amino acids 1–29, red) links the two monomers of the NS1 dimer. The wing domain (amino acids 30–180, yellow) of each monomer extends from the central β-roll. A discontinuous connector subdomain (amino acids 30–37 and 152–180, orange) is also linked to the dimerization domain. A major feature of NS1 is the third domain (amino acids 181–352, blue), with a ladder-like feature extending along the length of the dimer and comprised of 18 β-strands. Most of the inter-strand loops are short with the notable exception of a long “loop” (green) between amino acids 219–272 that lacks secondary structure but has extensive hydrogen bonding.

Previous studies have identified several immunodominent regions of the protein (Figure 1A,B). Immunodominent region 1 maps to the β-roll, immunodominant region 2 to a disordered sequence at the tip of the wing domain, immunodominent region 3 to the loop and immunodominant region 4 to the ends of the β-sheet ladder. Of these immuno-dominent regions 2 and 3 display the most sequence variability between flavivirus NS1 proteins. Site-directed mutagenesis was performed to generate mutated rNS1. Primers were designed to mutate residues in regions of NS1 with high sequence similarity to other members of the Flaviviridae family (Appendix A). Mutations were targeted toward sequences in the protein in immunodominant regions 2 and 3 that shared sequence identity with related flaviviruses. These regions were selected because they are the most highly conserved sequences in any of the immunodominant regions. Mutations of the sequence were introduced to the ZIKV NS1 sequence as triple alanine repeats to minimize impacts on folding of the protein structure. Specific mutations (117–119 and 227–229) were designed for lower reactivity of the protein to antibodies to non-ZIKV Flavivirus NS1 proteins and increase specificity of a diagnostic toward ZIKV.

### 3.2. Production of Mutant Zika Virus NS1 and Dengue Virus NS1

To produce recombinant non-structural protein 1 (rNS1) for the variant NS1, as well as the related DENV, sequences were amplified and cloned into the pET-45b(+) vector. The pET-45b(+) vector enables addition of an N-terminal His_6_ tag that was used for purification (Figure 2). After induction of protein production, multiple colonies were screened for expression by SDS-PAGE, with expression appearing to be near peak at 4 h post induction. Upon expression, localization was confirmed by sonication of an induced pellet followed by centrifugation to pellet insoluble mass and resuspension of that mass. This revealed the vast majority of protein to be in the insoluble fraction (inclusion body). Solubilization of the inclusion body was performed using urea-based buffers in order to denature the protein and pull it into solution. Following the protein through multiple purification steps shows the protein is not soluble in either the initial Triton X-100 solubilization buffer, or 1 M urea washes, but must be solubilized in 8 M urea. Ni-NTA purification of protein based on the added hexahistidine tag was performed on the 8 M urea soluble fraction with washes using 20 mM imidazole to eliminate non-specific binding. Elution was performed with 250 mM imidazole to outcompete the binding of the protein to the resin and allow the protein to flow through the column.

### 3.3. Reactivity of the rNS1 to Antibodies Circulating in Patient Serum and Immunized Goats

To compare the seroreactivity of the wild-type rNS1 to the mutant rNS1 samples from Zika patients from Columbia and the Dominican Republic were tested (Figure 3). IgG binding ratios for the Colombia samples show no differences between wild-type NS1 and either mutant protein. IgM binding ratios for the samples from the Columbian patients indicate increased binding to the wild-type protein. Samples from the Dominican Republic patients showed a different trend, with increased binding to the ZIKV NS1 117–119 and decreased binding to the ZIKV NS1 227–229. Both mutant NS1 proteins bound serum IgM more readily in the Dominican Republic samples compared to wild-type NS1 (Figure 3, Appendix A). Differences in protein–protein interactions or other structural changes in the mutant proteins could account for the differential antibody binding compared to wild-type NS1.

As a large volume of goat serum would be required to produce antigen-capture ELISA in sufficient amounts for the current and future studies sera from two goats immunized with either wild-type or mutant rNS1 were pooled. Differences in reactivity are present significantly in the pooled serum from the immunized goats. Goats immunized with the wild-type protein produce antibodies that bound to all three variants of ZIKV NS1, as well as DENV NS1, less strongly than antiserum produced from either mutant ZIKV NS1 proteins (Figure 4). The reduced reactivity, along with the data indicating high binding of the wild-type protein with antiserum from mutants, reinforces the idea that there are significant differences in the protein–protein interactions or structural differences of the mutate rNS1 compared to the wild-type protein. The mutant rNS1 bound each of the other proteins in amounts not significantly different, indicating that the mutations had little effect on binding of NS1, at least in goats.

### 3.4. Affinity Purification of ZIKV NS1 Polyclonal Antibodies

To generate a pAb solution that is specific to ZIKV NS1, without cross-reactivity to related viruses, we began with pooled goat anti-ZIKV NS1 serum generated from inoculation with the full-length ZIKV rNS1 117–119 mutant. This serum was selected because it had a higher titer for ZIKV rNS1 than the sera from goats immunized with wild-type rNS1. Using a CarboxyLink column coupled with the same mutant NS1, we purified anti-ZIKV pAbs via affinity chromatography. This resulted in a pool of antibodies with high levels of binding toward ZIKV NS1, but also possessed affinity for DENV2 NS1. This was removed via affinity chromatography using a similar column set-up but coupled with DENV2 NS1 wild-type protein (Figure 5A). This cross-adsorption was repeated once, resulting in a pAb solution with high affinity to ZIKV NS1 and low reactivity toward DENV2 NS1 (Figure 5B).

### 3.5. Development of a ZIKV NS1 Antigen-Capture ELISA

The anti-ZIKV NS1 pAbs generated were coated onto plates in order to perform antigen-capture ELISA targeted toward ZIKV NS1. Purified ZIKV NS1 W117A, G118A, K119A was diluted 1:4 in buffer beginning at 2000 ng/mL and going down to 0.49 ng/mL. This assay provided a good range of values corresponding with dilutions of NS1, with a limit of detection of between 1.95 and 7.8 ng/mL, defined as the mean of the negative control plus 1.96 standard deviation of the control (Figure 6A). Linear regression analysis shows a good fit of data, with an r^2^ value of 0.96 (Figure 6B). Comparison of separate runs using the same preparation of pAbs in both runs results in a *p* value < 0.0001 (Appendix A).

### 3.6. Specificity of ZIKV Antigen-Capture ELISA

Sequences of NS1 antigens from the related flaviviruses YFV, WNV and SLEV were generated synthetically and cloned into the pET-45b(+) vector by using primers that added appropriate restriction sites (Appendix A). As with our previous ZIKV and DENV NS1 constructs, this allowed for insertion of an N-terminal His_6_ sequence in order to allow for downstream blotting and purification. Protein staining of cultures induced with IPTG alongside uninduced colonies via Coomassie Blue reveals NS1 bands in sizes as expected (Figure 7). After solubilization of the NS1 proteins, which were located predominantly in the inclusion body, refolding was performed to return the proteins to their native structures.

Cross-reactivity of anti-ZIKV NS1 pAbs was analyzed via antigen-capture assay using ZIKV, DENV, SLEV, WNV and YFV rNS1. Binding values (A_450_) were significantly lower with concentrations of NS1 of ZIKV vs. DENV, SLEV, YFV and WNV from 2000 ng/mL to 7.8 ng/mL (Figure 8). West Nile Virus NS1 had cross-reactivity assessed via percentage of ZIKV NS1 binding at just above 20%, with all other NS1 proteins being at or below 20% cross-reactivity. Cross-reactivity of non-adsorbed antibodies was assessed as well. In this case, the cross-reactivity of WNV NS1 was over 70%, with other proteins falling between that and 50% (Appendix A). Non-cross-adsorbed antibodies react to ZIKV NS1 at approximately 55% of the level of cross-adsorbed antibodies in the same assay (Appendix A).

### 3.7. Antigen Levels in Serum from ZIKA Patients

We utilized the ZIKV NS1 antigen-capture assay to determine circulating NS1 levels in patient serum from the 2015–2016 ZIKV outbreak. Levels found circulating range from not detectable to over 2.2 µg/mL (Table 1). In the Colombian samples, 52.6% of samples were positive for NS1, while the Dominican Republic samples showed a higher positivity rate of 67.5%. Overall, 55.7% of samples collected from symptomatic ZIKV patients during the outbreak were positive (Table 1 and Appendix A).

We next asked if the detection of NS1 could be improved by dissociating the antigen from any circulating anti-Flavivirus NS1 antibodies patients may have developed from previous infections. If so, it would pose a challenge for detection of ZIKV infection in flavivirus endemic areas. Using an alkaline detergent method, we separated the antigen-antibody complex and performed NS1 antigen-capture ELISA as performed previously. This method showed little effect on detection of NS1 in solution (Appendix A). Performing the assay on samples from the Dominican Republic shows no significant change in detection of NS1 whether associated to circulating antibodies or not (Appendix A).

## 4. Discussion

The Zika epidemic of 2015–2016 spurred development of multiple antigen-capture assays [50,51,52]. Several have become commercially available for use in clinical diagnosis, while others were available for research use only [53]. Some of the diagnostics exhibited low limits of detection, and remained untested in the field [51], while others had sensitivities and specificities between 70% and 100% [54]. The broad sequence similarity of ZIKV proteins and other widespread flaviviruses, such as DENV, makes development of specific ZIKV diagnostic assays challenging. Generally, ZIKA antigen-capture assays have utilized mAbs to reduce reactivity to related flavivirus proteins. However, DENV diagnostics based on NS1 capture are susceptible to interference from antibodies circulating in patient serum, and other flavivirus diagnostics have exhibited the same issue. A diagnostic based on capture of WNV NS1 was developed using mAbs, where capture of recombinant NS1 was deemed sensitive, with a detection limit of 0.5 ng/mL soluble NS1. Detection in serum after day 7, even in mice during a primary infection, was decreased. This was presumably due to the increase in IgM generated against WNV NS1. After separation of the antigen–antibody complex, NS1 detection increased significantly [55]. This illustrates a potential difficulty with using mAb-capture NS1 diagnostics, which lead us to develop a test using pAbs. 

The hypothesis that mutations of NS1 that eliminate epitopes conserved among flaviviruses could decrease cross-reactivity to non-ZIKV flavivirus NS1 was tested. Region 2, containing an immunodominant region in both natural infection and vaccination, and mapped to the wing domain with a conserved amino acid sequence from aa 114–119 [56,57,58]. Monoclonal antibodies against this region are capable of inducing protection across multiple species within the *Flaviviridae* family [59]. We used NS1 with mutations in this epitope to analyze sera of serum collected from patients diagnosed with Zika from the 2015–2016 epidemics in Colombia and the Dominican Republic. Sera from the Columbian and Dominican human populations, previously diagnosed with Zika, reacted differently to the modified proteins compared to wild-type and the pattern in which these populations reacted differed. 

Immunizing with NS1 modified to distrupt cross-reactive epitopes significantly increased the immune response of goats relative to the wild-type protein. One possibility to explain this observation is that the mutations focused the animals’ humoral immune response to the remaining available non-cross-reactive epitopes. Our approach not only improved the specificity of the pAbs, but it also increased their apparent reactivity. This allowed the development of an NS1 antigen-capture ELISA that effectively measures soluble NS1 protein, with a limit of detection between 1.95 ng/mL and 7.8 ng/mL. This compares favorably to comparable DENV NS1 capture ELISAs, such as SD Bioline’s detection limit of between 16 and 63 ng/mL NS1 [60]. 

Future studies could potentially employ recombinant NS1 proteins with mutations in additional epitopes and combinations of different epitope mutations to further reduce cross-reactivity. For example, immunodominant region 4 in the NS1 protein corresponds to an area of NS1 above the beta-ladder that is exposed on the protein surface and is also an area of high antibody binding. This region elicited strong cross-reactive serological responses upon vaccination of mice with NS1 of DENV2 plus adjuvant, as well as during natural infection with DENV2. This points to this region, which is conserved among DENV strains, as an attractive target for the approaches piloted in the current study [58].

Expression of NS1 proteins of the genetically related flaviviruses YFV, WNV and SLEV allowed investigation of potential cross-reactivity that could generate false positive diagnostic results in our ZIKV antigen-capture ELISA. NS1 proteins from DENV, YFV, WNV and SLEV show minimal reactivity in our prototype ZIKV antigen-capture ELISA. The diagnostic revealed high ZIKV specificity, with the highest cross-reactivity coming from WNV, with only 20% of the signal generated from ZIKV NS1 at the highest level of protein tested. This is a decrease of 50% signal intensity from non-cross-adsorbed antibodies, indicating this process removes a substantial portion of cross-reactive antibodies from the solution, leaving a high signal generating, specific pool of pAbs. The ZIKV antigen-capture ELISA assay detected ZIKV in human serum from acute cases of ZIKV infection comparable to existing DENV antigen-capture diagnostics [61].

Cross reactivity is not the only problem in developing diagnostic tests where related viruses are endemic. Circulating anti-NS1 antibodies present during later in infection can reduce assays sensitivity dramatically [42,49]. The pAb-based ELISA showed no effect of antigen unmasking, indicating that this phenomenon is not a concern as it can be with mAb-based assays. Not only does the pAb approach contribute to solving the need for more ZIKV-specific assays that must be in place to differentiate ZIKV infections from similar and related infections, but none of the masking of NS1 by pre-existing antibodies generated by previous infections by related flaviviruses was observed. These results also suggest that a rapid diagnostic test could be developed using this approach to satisfy a need for point of care diagnostics for deployment to sites of outbreaks to be performed by personnel without extensive training [62]. This could also be utilized in a side-by-side antigen and antibody (IgM) detection platform, as is available for DENV infection [42,63].

Evaluation of different methods of generating ZIKV-specific antibodies and antigens may be fruitful in the future. Rabbits are commonly used to generate antibodies, and rabbit antibodies often possess enhanced binding capabilities [64,65]. Use of rabbit antibodies in NS1 antigen-capture may require less cross-adsorption, as well as resulting in low non-specific interactions during lateral flow rapid tests. Generation of protein via mammalian cells may be of benefit as well, as mammalian cells can produce post-translational modifications, including glycosylation, that result in an antigen with greater similarity to that produced during human infection, thereby improving antibody recognition. Advances in mammalian cell-hosted protein production [66,67] suggest that yields sufficient for the implementation of the type of detection test presented in this report are achievable.

## 5. Conclusions

An antigen-capture ELISA configured with an affinity-purified pAb to ZIKV NS1 demonstrated limited cross-reactivity to NS1 of commonly circulating flaviviruses. Refinement of approaches similar to those employed here could lead to development of ZIKV-specific immunoassays suitable for use in areas where infections with related flaviviruses are common. This assay is effective at detection of ZIKV NS1 even when antigens from ZIKV or co-circulating non-ZIKV flavivirus antibodies are present. These results further the goal of developing the tools for ZIKV infection diagnoses at every window of infection, thus giving the clinician valuable information for future case management and treatment. 

## Figures and Tables

**Figure 1 viruses-13-01771-f001:**
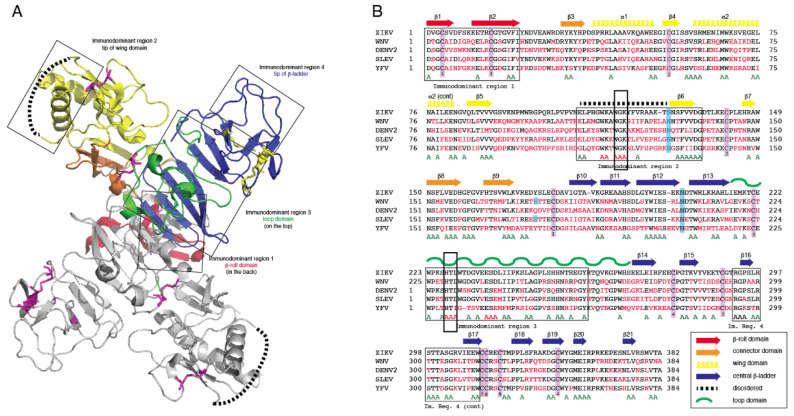
Comparison of non-structural protein 1 sequences among closely related flaviviruses. (**A**): Ribbon model highlighting regions of the NS1 protein containing segments exposed at the outer surface to the host environment. (**B**): Sequence comparison showing regions with high sequence disparity. Amino acids depicted in red differ from the corresponding ZIKV NS1 amino acids. A represents positions with two sequences with amino acids identical to ZIKV NS1. The boxes highlight highly conserved sequences, amino acids 117–119 and 227–229, that were mutated to alanine in immunodominant regions 2 and 3.

**Figure 2 viruses-13-01771-f002:**
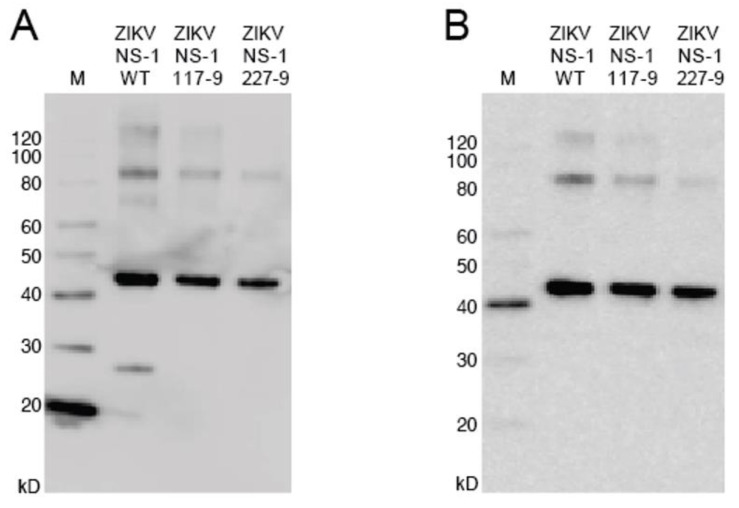
Western blot of NS1 mutants. (**A**): The blot was probed with anti-His6 antibody targeted toward the protein N-terminus. (**B**): The blot was probed with anti-ZIKV NS1 monoclonal antibody targeted toward the C-terminus. Uncropped gels are displayed in Appendix A.

**Figure 3 viruses-13-01771-f003:**
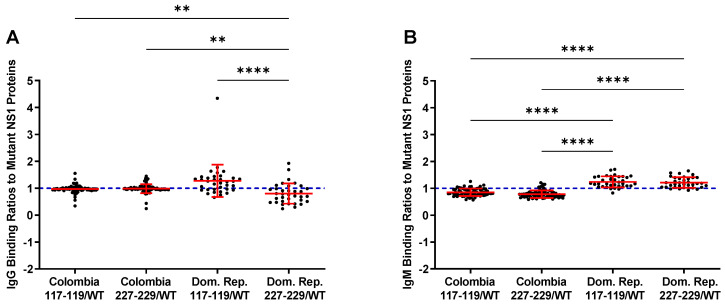
Ratios of mutant vs. wild-type binding of NS1 by patient samples. The blue line at 1 indicates equal binding. Numbers over 1 indicate increased binding to the named mutant, while under 1 indicate increased wild-type binding. (**A**): Ratios of mutant/wild-type binding from IgG in samples show preferential binding to 117–119 mutant NS1. (**B**): IgM shows preferential binding of mutant NS1 proteins in samples from the Dominican Republic. Colombia = Colombia samples from suspected ZIKV infection. Dom Rep = Dominican Republic samples from suspected ZIKV infection. 117–119 = ZIKV NS1 W117A, G118A, K119A. 227–229 = ZIKV NS1 H227A, T228A, L229A Comparisons were made using Kruskal–Wallis ANOVA. Asterisks represent significant comparisons (**** *p* < 0.0001; ** *p* < 0.01). The full dataset is presented in Appendix A.

**Figure 4 viruses-13-01771-f004:**
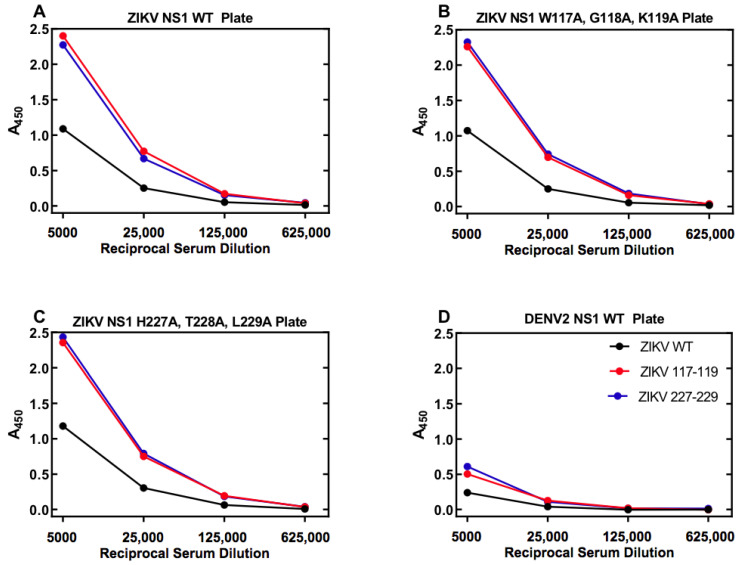
Serum from goats immunized with ZIKV mutants show differential binding to NS1 proteins. Antibody-capture ELISA was performed on a mixture of serum from two goats immunized with ZIKV NS1 protein. Binding of WT ZIKV NS1 showed lower binding to WT (**A**), both mutants (**B**,**C**), as well as DENV2 WT NS1 (**D**). Data represent samples run in duplicate. Error bars (standard error of the mean) were smaller than the symbols as drawn. The experiment was repeated a total of three times. The legend in panel (**D**) also applies to panels (**A**–**C**).

**Figure 5 viruses-13-01771-f005:**
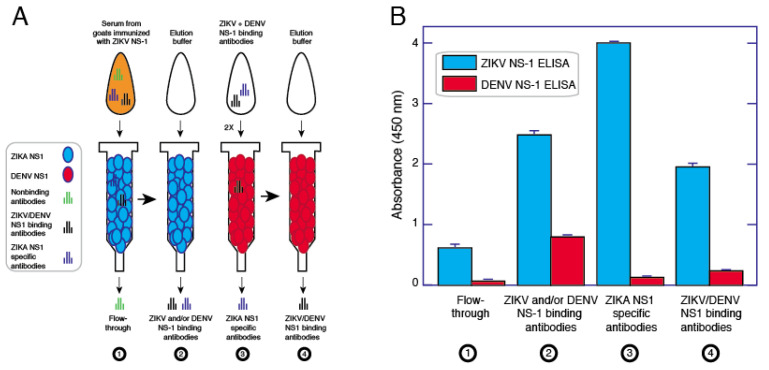
Process for purifying and cross-adsorbing ZIKV NS1 antibodies. ZIKV NS1 antibodies were affinity purified using a gravity flow column. (**A**): The bound and eluted fraction was put through a column with DENV NS1 twice. The unbound and eluted (non-DENV reactive) fraction is the cross-adsorbed, ZIKV-specific fraction. (**B**): Antibodies flowing through the column initially bind ZIKV at low signal strength. Following ZIKV NS1 column elution, strength of binding of the solution goes up, but DENV NS1 reactivity is present. Upon cross-adsorption against DENV NS1, specificity of the pAb solution goes up, as well as its binding avidity to ZIKV NS1.

**Figure 6 viruses-13-01771-f006:**
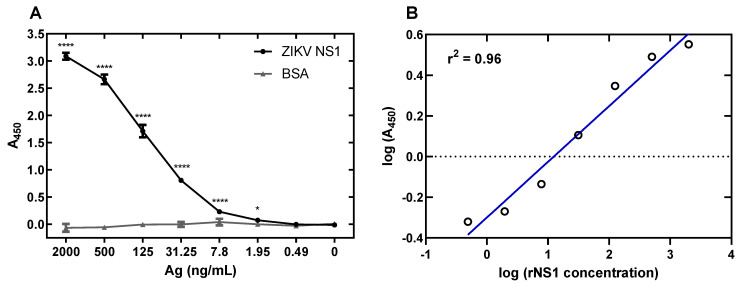
ZIKV NS1 antigen-capture ELISA limit of detection and dynamic range. (**A**): ZIKV rNS1 shows binding significantly higher than BSA down to 7.8 ng/mL. The limit of detection of the assay is between 7.8 and 1.95 ng/mL NS1. (**B**): Linear regression of NS1 detection via ELISA demonstrates assay can quantify protein across a range of values. Comparisons were made using two-way ANOVA with a Holm–Sidak multiple comparisons test. Asterisks represent significant comparisons (**** *p* < 0.0001; * *p* < 0.05). Error bars represent the standard error of the mean.

**Figure 7 viruses-13-01771-f007:**
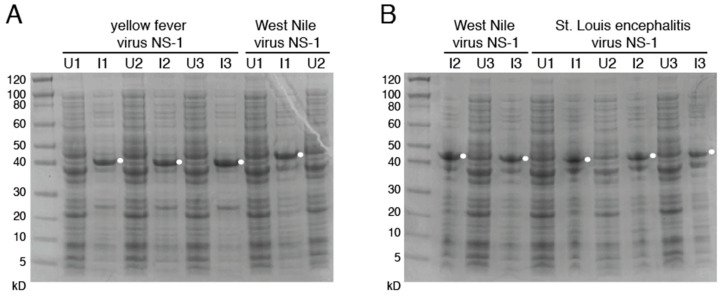
Production of flavivirus NS1. Coomassie-stained SDS-PAGE images of induced (I) and uninduced (U) cultures from NS1 proteins of yellow fever virus (**A**), West Nile virus (**A**,**B**) and St. Louis Encephalitis virus (**B**). Bands indicate proteins similar to DENV and ZIKV NS1 WT production. Results from three different colonies [1,2,3] for each flavivirus NS1 are indicated. Uncropped gels are displayed in Appendix A.

**Figure 8 viruses-13-01771-f008:**
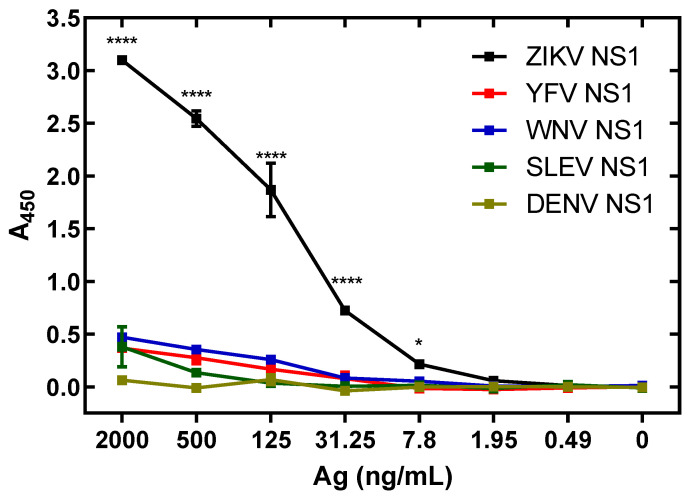
The ZIKV NS1 antigen-capture assay demonstrates low cross-reactivity to related viruses. Recombinant NS1 antigens were produced and assayed. A: Even at low levels of NS1 detection, the assay is specific to ZIKV NS1. Comparisons made using two-way ANOVA with a Holm–Sidak multiple comparisons test. Asterisks represent significant comparisons between ZIKV NS1 and all other NS1 proteins (**** *p* < 0.0001; * *p* < 0.05). Error bars represent the standard error of the mean.

**Table 1 viruses-13-01771-t001:** ZIKV NS1 antigen-capture ELISA results from patients infected during the 2015–2016 outbreaks in Colombia and the Dominican Republic ^1^.

Country	Number of Samples	Positive Samples (%)
Columbia	154	81 (52.6)
Dominican Republic	40	27 (67.5)
Overall	194	108 (55.7)

^1^ Raw data for this Table are presented in Appendix A.

## Data Availability

Not applicable.

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
