# Peer review of "Zika Virus Non-Structural Protein 1 Antigen-Capture Immunoassay"

_viruses, 2021, doi:10.3390/v13091771_

Round 1
Reviewer 1 Report
In the manuscript by Beddingfield et al. the objectives of the study are well articulated and the study design seems appropriate to respond to the research question. Most of the results are convincing but not really well exposed in the manuscript. In particular, the graphs of Figures 1-6 are not at the right places in the main text which complicates the reading of the manuscript. More importantly, it is not clear if the protein refolding step which is based on the use of gluthatione were performed on recombinant Zika and dengue NS1 proteins in the Results section 3.2 as the other flavivirus NS1 antigens in section 3.6. The authors need to clarify if the purified recombinant Zika and Dengue NS1 antigens do exist as dimers or only as monomers. A lack of oligomerization of recombinant NS1 protein could involve that most of antibody epitopes exist as linear B-cell epitopes. Could this explain that NS1 antigen unmasking assay lacks of effect by ELISA method using goat anti-Zika NS1 immune serum ? There are additional points that should be taken in account to provide more clarity.
- Lines 87-88: Introduction; the section could include information on the persistence of Zika virus in human body fluids and the possible implication for viral diagnosis.
- Materials and Methods, the section does provide any information on the immunoblot assay presented in Figure 2.
- Lines 111-13: Materials and Methods; the authors should include details on the recombinant Zika NS1 protein (viral strain origin, genotype, Genbank number access, codon usage allowing expression in bacteria, …) used in the study.
- Lines 123-24: Materials and Methods; it is not clear if the sequencing of plasmids encoding ZIKV NS1 and mutants was performed.
- Lines 191-92: Materials and Methods; it is not clear which recombinant NS1 proteins were inoculated in goats.
- Lines 233-34-92: Materials and Methods; the authors should include details on the biotinylated-NS1 pAbs used in their antigen-capture ELISA.
- Lines 282-84: Figure 2B; the authors should include details on the anti-ZIKV NS1 mAb used in their study.
- Line 314: Results 3.2; it is not clear in which the Figure 2 demonstrates the purification of recombinant antigen using His-tag.
- Lines 334-339. Results 3.3. The paragraph is not clear for the reader.
- Figure 4. The graphs A-C are lacking of legends for the three colored lines. It is not clear why a pool of serum from two immunized goats has been used for ELSIA despite a lack of information on the individual antibody response of each animal to recombinant antigen. The figure legend is not clear for the reader.
- Figure 7. Legend. The numbers 1-3 associated to I or U correspond to what ?
- Table S3. The authors should confirm a direct link between serological data listed in Table S3 and those presented in Figure 3 as well as Table 1.
Minor point:
Line 286: “proteins” but not “protiens”
Author Response
Comments and Suggestions for Authors – Reviewer 1
In the manuscript by Beddingfield et al. the objectives of the study are well articulated and the study design seems appropriate to respond to the research question.
We appreciate the positive comments of this reviewer as well as the detailed and very helpful guidance. This has allowed us to greatly improve the presentation and clarity of the manuscript.
Most of the results are convincing but not really well exposed in the manuscript. In particular, the graphs of Figures 1-6 are not at the right places in the main text which complicates the reading of the manuscript.
We have properly placed the figures in the revised manuscript.
More importantly, it is not clear if the protein refolding step which is based on the use of gluthatione were performed on recombinant Zika and dengue NS1 proteins in the Results section 3.2 as the other flavivirus NS1 antigens in section 3.6.
All of the rNS1 Proteins used were refolded as indicated in revision.
The authors need to clarify if the purified recombinant Zika and Dengue NS1 antigens do exist as dimers or only as monomers. A lack of oligomerization of recombinant NS1 protein could involve that most of antibody epitopes exist as linear B-cell epitopes. Could this explain that NS1 antigen unmasking assay lacks of effect by ELISA method using goat anti-Zika NS1 immune serum ?
Size exclusion chromatography indicated that the wild-type and mutant NS1 proteins assemble as dimers. Residual dimers are present in the SDS gels. We have added a discussion of this point as suggested by the reviewer.
There are additional points that should be taken in account to provide more clarity.
- Lines 87-88: Introduction; the section could include information on the persistence of Zika virus in human body fluids and the possible implication for viral diagnosis.
We agree that ZIKV persistence is important to include as background and have referenced this fact in the revised manuscript.
- Materials and Methods, the section does provide any information on the immunoblot assay presented in Figure 2.
The immunoblot method was included in revision.
- Lines 111-13: Materials and Methods; the authors should include details on the recombinant Zika NS1 protein (viral strain origin, genotype, Genbank number access, codon usage allowing expression in bacteria, …) used in the study.
We added these details to the Methods. Rosetta2 host strains are designed to enhance the expression of eukaryotic proteins that contain codons rarely used in E. coli. These strains supply tRNAs driven by their native promoters for 7 rare codons on a chloramphenicol-resistant plasmid.
- Lines 123-24: Materials and Methods; it is not clear if the sequencing of plasmids encoding ZIKV NS1 and mutants was performed.
All plasmids were sequenced, which is indicated in revision.
- Lines 191-92: Materials and Methods; it is not clear which recombinant NS1 proteins were inoculated in goats.
We clarify in revision that wild-type NS1 and NS1 with mutations (117-119 and 227-229) were each inoculated in goats.
- Lines 233-34-92: Materials and Methods; the authors should include details on the biotinylated-NS1 pAbs used in their antigen-capture ELISA.
Additional details are now included about the biotinylated-NS1 pAbs used in the antigen-capture ELISA.
- Lines 282-84: Figure 2B; the authors should include details on the anti-ZIKV NS1 mAb used in their study.
The revised Methods section indicated that this Mab was purchased from a commercial source (GeneTex).
- Line 314: Results 3.2; it is not clear in which the Figure 2 demonstrates the purification of recombinant antigen using His-tag.
This has been clarified.
- Lines 334-339. Results 3.3. The paragraph is not clear for the reader.
This paragraph has been rewritten for accuracy and clarity.
- Figure 4. The graphs A-C are lacking of legends for the three colored lines.
We now indicate that the legend applies to all panels.
It is not clear why a pool of serum from two immunized goats has been used for ELSIA despite a lack of information on the individual antibody response of each animal to recombinant antigen.
We have clarified that pooling was performed to increase the amount of serum available for the present and future analyses.
The figure legend is not clear for the reader.
The figure legend has been rewritten for accuracy and clarity.
- Figure 7. Legend. The numbers 1-3 associated to I or U correspond to what ?
These numbers refer to separate analysis of three different colonies.
- Table S3. The authors should confirm a direct link between serological data listed in Table S3 and those presented in Figure 3 as well as Table 1.
We added the appropriate links in several places ion the revised manuscript.
Minor point:
Line 286: “proteins” but not “protiens”
We corrected his typographical error.
Reviewer 2 Report
Beddingfield et al develop an NS1 ELISA with increased specificity towards Zika virus by both altering conserved epitopes in Zika NS1 and depleting antibodies with high affinity to Dengue NS1. Figure 3 was not present in the pdf, which prevented thorough critic of the manuscript data.
Major:
- Figure 1, can you add what the “As” below the sequence are referencing. Is the color significant? Only two mutant NS1 variants were created, was there any logic behind choosing the two locations?
- Figure 2 is not referred to in the text.
- Figure 3 was not present in the pdf and therefore I could not evaluate it.
- Figure 4 “was run in duplicate and repeated for a total of three runs”, can error bars be added?
- Figure 5 – This used the ZIKV antibodies from goats immunized with the 117-119AAA NS1. If you compared this method with wt NS1 antibodies it may strength the argument that the mutation helped increase ZIKV specificity.
Minor:
- Intro paragraph 3 reports rather high symptoms rates for Zika infection. Are these referring to “symptomatic cases”? May want to clarify the denominator in these statements.
- The degree symbol is wrong throughout the manuscript.
- Line 405-407 – the sentence is confusing, suggest rewording.
Author Response
Comments and Suggestions for Authors – Reviewer 2
Beddingfield et al develop an NS1 ELISA with increased specificity towards Zika virus by both altering conserved epitopes in Zika NS1 and depleting antibodies with high affinity to Dengue NS1. Figure 3 was not present in the pdf, which prevented thorough critic of the manuscript data.
We thank Reviewer 2 for extremely helpful comments. Figure 3 is present in the revised pdf.
Major:
- Figure 1, can you add what the “As” below the sequence are referencing. Is the color significant?
Amino acids depicted in red differ from the corresponding ZIKV NS1 amino acids. A represents positions with two sequences with amino acids identical to ZIKV NS1.
Only two mutant NS1 variants were created, was there any logic behind choosing the two locations?
Those are the most highly conserved sequences in any of the immunodominant regions. This point is added in revision.
- Figure 2 is not referred to in the text.
We corrected this error.
- Figure 3 was not present in the pdf and therefore I could not evaluate it.
Figure 3 is present in the revised pdf.
- Figure 4 “was run in duplicate and repeated for a total of three runs”, can error bars be added?
We have clarified this point. The error bars in this figure were smaller than the size of the symbols.
- Figure 5 – This used the ZIKV antibodies from goats immunized with the 117-119AAA NS1. If you compared this method with wt NS1 antibodies it may strength the argument that the mutation helped increase ZIKV specificity.
The titer of serum from the goats immunized with wt NS1 (Fig. 4) was too low to justify affinity purification.
Minor:
- Intro paragraph 3 reports rather high symptoms rates for Zika infection. Are these referring to “symptomatic cases”? May want to clarify the denominator in these statements.
We clarified this statement as suggested.
- The degree symbol is wrong throughout the manuscript.
We corrected this.
- Line 405-407 – the sentence is confusing, suggest rewording.
We reworded this sentence for clarity.
Round 2
Reviewer 2 Report
Changes were sufficient to address concerns